# Structural insights into the molecular mechanism of the m$^6$A writer complex

**Paweł Śledź, Martin Jinek\***

Department of Biochemistry, University of Zurich, Zurich, Switzerland

**Abstract** Methylation of adenosines at the N$^6$ position (m$^6$A) is a dynamic and abundant epitranscriptomic mark that regulates critical aspects of eukaryotic RNA metabolism in numerous biological processes. The RNA methyltransferases METTL3 and METTL14 are components of a multisubunit m$^6$A writer complex whose enzymatic activity is substantially higher than the activities of METTL3 or METTL14 alone. The molecular mechanism underpinning this synergistic effect is poorly understood. Here we report the crystal structure of the catalytic core of the human m$^6$A writer complex comprising METTL3 and METTL14. The structure reveals the heterodimeric architecture of the complex and donor substrate binding by METTL3. Structure-guided mutagenesis indicates that METTL3 is the catalytic subunit of the complex, whereas METTL14 has a degenerate active site and plays non-catalytic roles in maintaining complex integrity and substrate RNA binding. These studies illuminate the molecular mechanism and evolutionary history of eukaryotic m$^6$A modification in post-transcriptional genome regulation.

## Introduction

Methylation of the amino group at the N$^6$ position in adenine nucleobases (m$^6$A) has emerged as a dynamic posttranscriptional RNA modification (*Fu et al., 2014*; *Liu and Pan, 2016*; *Meyer and Jaffrey, 2014*; *Yue et al., 2015*). As the most abundant internal modification in eukaryotic messenger and long non-coding RNAs (*Dominissini et al., 2013*; *Meyer et al., 2012*), m$^6$A has been shown to regulate the processing (*Alarcón et al., 2015b*; *Xiao et al., 2016*), translation (*Lin et al., 2016*; *Wang et al., 2015*) and stability (*Wang et al., 2014a*) of cellular transcripts. In mRNAs, the modification is typically found clustered around the stop codon and within 3′ untranslated regions (*Dominissini et al., 2012*; *Meyer et al., 2012*); the presence of m$^6$A in 5′ untranslated regions has in turn been shown to promote 5′ cap-independent translation (*Meyer et al., 2015*). m$^6$A is implicated in controlling stem cell differentiation and embryonic development (*Batista et al., 2014*; *Chen et al., 2015*; *Geula et al., 2015*; *Wang et al., 2014b*), the mammalian circadian clock (*Fustin et al., 2013*), as well as in stress responses such as heat shock (*Zhou et al., 2015*). The modification is reversible due to the interplay of methyltransferase (MTase) 'writer' (*Liu et al., 2014*) and demethylase 'eraser' (*Jia et al., 2011*; *Zheng et al., 2013*) enzymes. Transcripts containing m$^6$A marks are specifically recognized and regulated by 'reader' proteins, in particular members of the YTH domain family (*Stoilov et al., 2002*). The cytoplasmic protein YTHDF2 specifically recognizes and destabilizes m$^6$A-modified RNAs by promoting their localization to processing (P-) bodies (*Wang et al., 2014a*), while YTHDF1 stimulates mRNA translation by interactions with translation initiation factors (*Wang et al., 2015*). The nuclear YTHDC1 protein modulates pre-mRNA splicing by recruiting and modulating splicing factors (*Xiao et al., 2016*). Another m$^6$A reader, HNRNPA2B1, has been shown to promote pri-microRNA processing through interactions with the Microprocessor complex (*Alarcón et al., 2015a*).

The m$^6$A RNA modification is catalyzed in the nucleus by a multiprotein writer complex containing the putative methyltransferases METTL3 and METTL14 (*Bokar et al., 1997*; *Liu et al., 2014*;

**\*For correspondence:** jinek@bioc.uzh.ch

**Competing interests:** The authors declare that no competing interests exist.

*Schwartz et al., 2014*). The two subunits interact directly with each other and further associate with Wilms tumor 1-associated protein (WTAP) (*Liu et al., 2014*; *Ping et al., 2014*; *Schwartz et al., 2014*). WTAP does not have MTase activity but is required for efficient RNA methylation in vivo and for localization of METTL3 and METTL14 to nuclear speckles (*Ping et al., 2014*). The m[6]A writer complex catalyzes the transfer of a methyl group from the donor substrate S-adenosyl methionine (SAM) to the adenine nucleobases in acceptor RNA substrates (*Bokar et al., 1997*). The complex preferentially modifies substrate RNAs conforming to the consensus sequence GGACU but otherwise appears to have little specificity for structural features in its RNA substrates (*Liu et al., 2014*). Both METTL3 and METTL14 have been shown to have weak m[6]A methyltransferase activity in isolation. However, physical interaction between METTL3 and METTL14 results in substantially higher MTase activity than the sum of the activities of the individual methyltransferase subunits (*Liu et al., 2014*). Despite recent advances in understanding the biological roles of m[6]A modification, the molecular architecture of the m6A writer complex and its mechanism of RNA methylation are poorly understood. In particular, the molecular basis for the synergistic enhancement of MTase activity within the m[6]A writer complex is unknown.

To shed light on the molecular mechanism of m[6]A RNA methylation, we determined crystal structures of the catalytic core of the m[6]A writer complex comprising METTL3 and METTL14 methyltransferase subunits. The structures reveal the molecular architecture of the complex and highlight specific features involved in complex assembly, substrate recognition and catalysis. Crucially, our structural and biochemical data imply that METTL3 is the sole catalytic subunit of the complex, while METTL14 plays non-catalytic roles in complex stabilization and substrate RNA recruitment. These findings advance our mechanistic understanding of the m[6]A modification in post-transcriptional genome regulation.

## Results

### Identification and structure determination of a minimal core m[6]A writer complex

The m[6]A writer complex contains two putative catalytic subunits - METTL3 and METTL14. Previous studies have shown that METTL3 and METTL14 physically interact with each other and that their association has a synergistic effect on the catalytic activity of the complex (*Liu et al., 2014*). To obtain structural insights into the molecular architecture and mechanism of the m[6]A writer complex, we sought to determine the crystal structure of the catalytic core of the complex comprising the two methyltransferase (MTase) subunits (*Figure 1A*). To this end, we co-expressed human METTL3 and METTL14 in insect cells. The purified METTL3-METTL14 complex was catalytically active (*Figure 1B*) but failed to crystallize. Both METTL3 and METTL14 polypeptides contain a central MTA-70 methyltransferase domain (*Bujnicki et al., 2002*). METTL3 additionally contains an N-terminal region harboring two putative CCCH zinc finger motifs, whereas in METTL14, the MTA-70 domain is flanked by N- and C-terminal extensions containing low-complexity sequences (*Iyer et al., 2016*) (*Figure 1A*). To identify protein constructs amenable to crystallization, the full-length METTL3-METTL14 complex was subjected to limited proteolysis with chymotrypsin and N-terminal sequencing. Based on these experiments, we engineered N- and C-terminally truncated forms of both METTL3 and METTL14, comprising residues 354–580 and 107–395, respectively. The MTase activity of the truncated METTL3-METTL14 dimer was reduced to background level, indicating that the terminal extensions are required for efficient substrate RNA binding or catalysis. Further experiments using complexes in which only METTL3 or METTL14 was truncated showed that the catalytic activity is contingent upon the presence of full-length METTL3, while the N- and C-terminal extensions flanking the central region comprising residues 107–395 in METTL14 are dispensable (*Figure 1B*). The truncated METTL3-METTL14 complex was crystallized and its structure solved by a single-wavelength anomalous diffraction approach utilizing the anomalous signal of native sulfur atoms. Experimentally-phased electron density maps revealed the presence of a single molecule of the donor product S-adenosyl homocysteine (SAH), serendipitously co-purified with the complex, in the METTL3 active site (*Figure 1—figure supplement 1A*). High-resolution native data were subsequently measured from a METTL3-METTL14 crystal grown in the presence of SAH and a short m[6]A-containing oligonucleotide. However, the RNA oligonucleotide was not evident in electron density

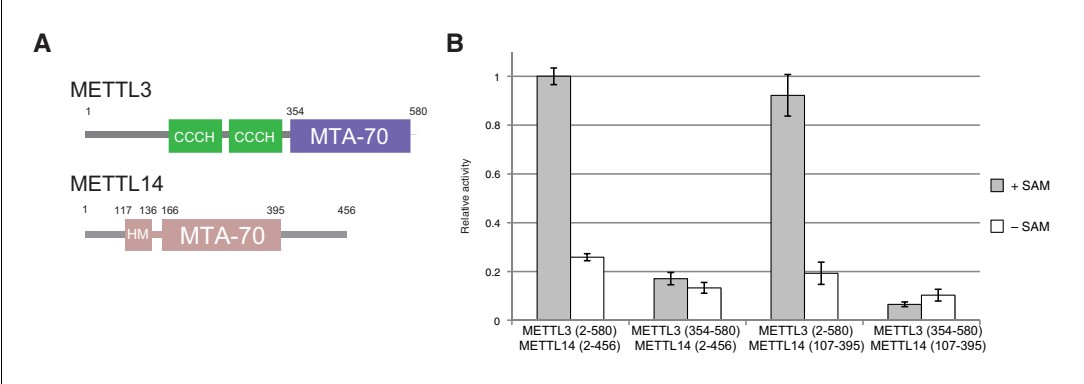

**Figure 1.** Domain composition and methyltransferase activity of METTL3-METTL14. (**A**) Domain composition of METTL3 and METTL14. Predicted CCCH zinc finger motifs in the N-terminal region of METTL3 are shown in green. The crystallized METTL3-METTL14 complex contained the MTA-70 methyltransferase domain of METTL3 (residues 354–580, highlighted in blue) and a METTL14 construct (residues 117–395, highlighted in pink) comprising the MTA-70 domain preceded by an N-terminal extension (denoted HM). (**B**) Methyltransferase activities of the full-length METTL3-METTL14 complex, and complexes containing truncated METTL3 and METTL14 proteins used for the structure determination. The activities were determined by measuring the concentration of methylated RNA product using a m⁶A-specific antibody in an enzyme-linked immunosorbent assay. The activities are represented as fractions of the activity of WT complex. Reactions lacking the donor substrate SAM were performed to determine background signal in the assay due to non-specific antibody binding. Error bars represent mean ± standard error of the mean (s.e.m) of three independent replicates.

The following figure supplements are available for figure 1:

**Figure supplement 1.** Methyl group donor substrate and product binding in the METTL3 active site.

**Figure supplement 2.** METTL3-METTL14 is catalytically active under reducing conditions.

maps. The final atomic model, containing a single METTL3-METTL14 dimer in the asymmetric unit, was refined at a resolution of 1.85 Å with an $R_{free}$ of 19.0%, working $R$ of 15.7%, and good stereo-chemistry (*Table 1*). We additionally co-crystallized the METTL3-METTL14 core complex in the presence of the donor substrate S-adenosyl methionine (SAM) (*Figure 1—figure supplement 1B*). Using these crystals, a structure of the METTL3-METTL14 complex, in which the co-purified SAH product was exchanged by SAM, was determined at a resolution of 1.9 Å and refined to an $R_{free}$ and $R_{work}$ values of 18.6% and 15.7%, respectively. No additional electron density attributable to SAM or SAH could be observed in the proximity of METTL14 in the two structures. Disulfide bridge formation was observed between Cys338[METTL14] and Cys388[METTL14] (*Figure 1—figure supplement 2A*). The MTase activity of the METTL3-14 complex has previously been reported to be sensitive to reducing agents (*Li et al., 2016*), suggesting that disulfide bond formation could play a role in regulating the catalytic activity of the complex. However, substitution of Cys338[METTL14] with alanine had little impact on MTase activity, indicating that disulfide bond formation in METTL14 is not required for the function of the m⁶A writer complex (*Figure 1—figure supplement 2B*).

## METTL3 and METTL14 form a pseudosymmetric dimer

The structures of the METTL3-METTL14 complex reveal that the two methyltransferase subunits form a compact pseudosymmetric heterodimer with overall dimensions of 80 × 40 × 35 Å (*Figure 2A*). Both METTL3 and METTL14 belong to the Group I clade of eukaryotic N⁶A-MTases (*Iyer et al., 2016*). The core of each methyltransferase subunit is formed by a Rossman fold domain comprising a central, curved, eight-stranded beta sheet flanked by four alpha helices (*Figure 2B*, *Figure 2—figure supplement 1*). The METTL3 and METTL14 MTase domains dimerize through an extensive interface that buries 2055 Å² of solvent-accessible surface area per subunit. The symmetric part of the dimerization interface is formed by strands β4–5 and helix α4 of each MTase domain (*Figure 2C*), and involves a network of hydrogen bonding interactions as well as hydrophobic contacts (*Figure 2D*). The hydrophobic region of the METT3-METTL14 interface, centered on strands

**Table 1.** Data collection and refinement statistics.

| | SAM complex | SAH complex | Sulfur-SAD |
|---|---|---|---|
| X-ray source | SLS X06DA (PXIII) | SLS X06DA (PXIII) | SLS X06DA (PXIII) |
| Space group | $P\,3_2\,2\,1$ | $P\,3_2\,2\,1$ | $P\,3_2\,2\,1$ |
| Cell dimensions | | | |
| $a, b, c$ (Å) | 63.95, 63.95, 225.86 | 63.82, 63.82, 225.63 | 64.02, 64.02, 226.84 |
| $\alpha, \beta, \gamma$ (°) | 90.0, 90.0, 120.0 | 90.0, 90.0, 120.0 | 90.0, 90.0, 120.0 |
| Wavelength (Å) | 1.03965 | 1.03965 | 2.07334 |
| Resolution (Å) | (2.02–1.90) | (1.96–1.85) | (2.80–2.70) |
| $R_{sym}$ (%) | 8.9 (70.6) | 7.5 (68.9) | 16.5 (97.3) |
| $CC_{1/2}$ | 99.9 (89.0) | 100.0 (87.5) | 100.0 (97.9) |
| $I\,/\,\sigma I$ | 21.89 (3.42) | 24.51 (3.55) | 47.24 (8.03) |
| Observations | 846792 (131654) | 906619 (142053) | 3073142 (279442) |
| Unique reflections | 81250 (13087) | 87473 (14080) | 28625 (2970) |
| Multiplicity | 10.4 (10.1) | 10.4 (10.1) | 107.4 (94.1) |
| Completeness (%) | 99.9 (99.2) | 99.9 (99.3) | 100.0 (100.0) |
| **Refinement** | | | |
| Resolution (Å) | 44.62–1.90 | 49.65–1.85 | |
| No. reflections | 43271 | 46510 | |
| $R_{work}/R_{free}$ | 0.1567/0.1855 | 0.1569/0.1899 | |
| **No. atoms** | | | |
| Protein | 3549 | 3571 | |
| Ligands | 32 | 31 | |
| Water | 359 | 405 | |
| **B-factors** | | | |
| Mean | 28.76 | 29.89 | |
| Protein | 28.17 | 29.08 | |
| Ligands | 25.06 | 22.72 | |
| Water | 35.28 | 37.64 | |
| **R.m.s. deviations** | | | |
| Bond lengths (Å) | 0.011 | 0.011 | |
| Bond angles (°) | 1.088 | 1.126 | |
| Ramachandran plot | | | |
| % Favored | 98.1 | 98.8 | |
| % Allowed | 1.9 | 1.2 | |
| % Outliers | 0 | 0 | |

Values in parentheses denote highest resolution shell.

β4–5, is capped by the side chain of Ile134[METTL14] from an N-terminal extension of METTL14 comprising residues Asn117-Asp136[METTL14] (*Figure 1A*). The METTL14 N-terminal extension, which lacks secondary structure except for a short alpha-helical motif (Tyr119-Thr126[METTL14]), breaks the overall pseudo-C2 symmetry of the heterodimer and shields the hydrophobic dimerization interface from solvent exposure. Of note, we were unable to express soluble METTL3-METTL14 complexes lacking the N-terminal METTL14 extension. Unlike METTL14, METTL3 lacks equivalent amino acid residues in the region upstream of the methyltransferase domain. Finally, the dimerization interaction is reinforced by two partially disordered loops, one from each subunit (residues Arg468-Lys480[METTL3] and

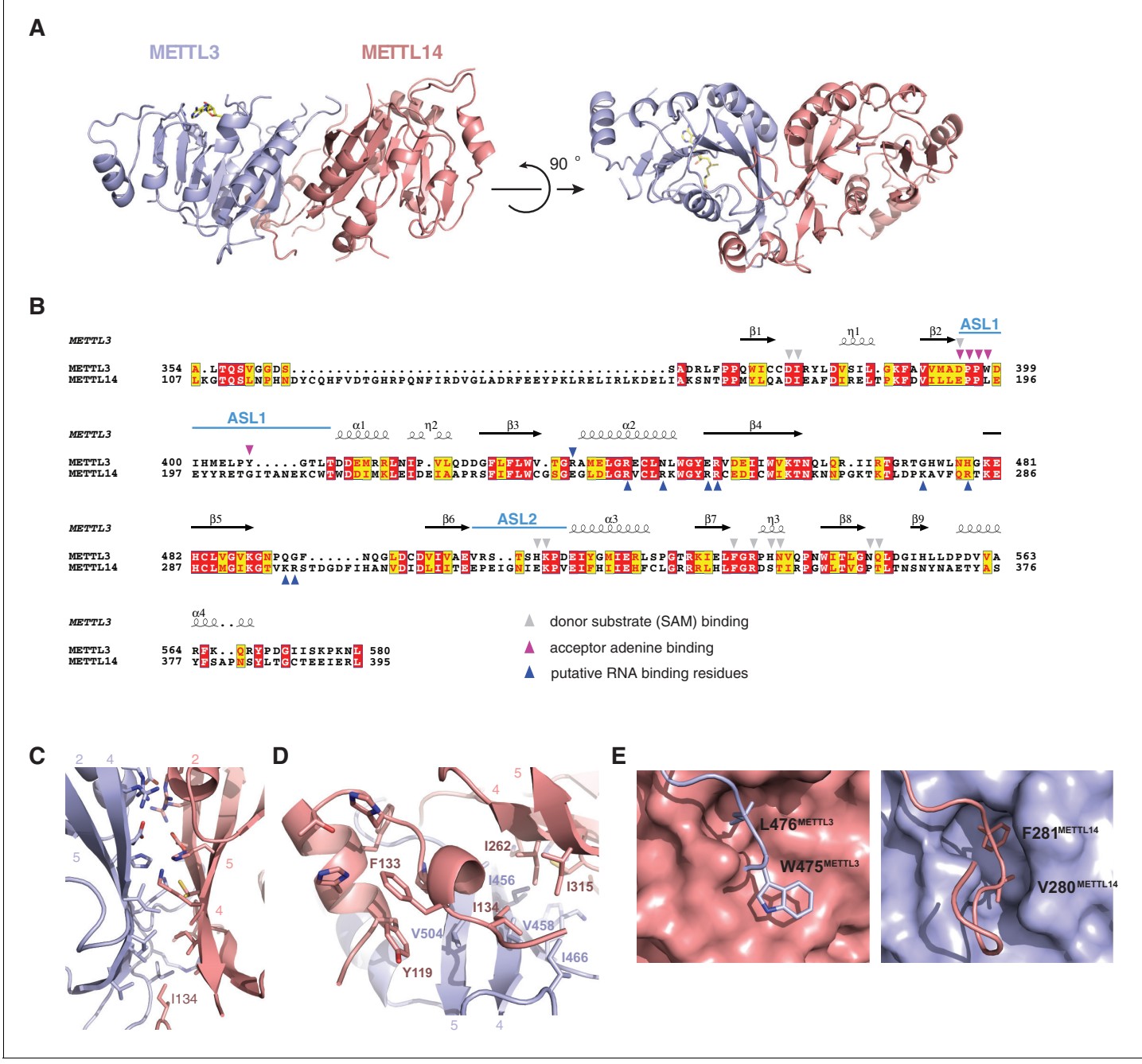

**Figure 2.** METTL3 and METTL14 form an asymmetric dimer. (**A**) Overall structure of METTL3-METTL14 heterodimer. METT3 is colored blue; METTL14 is colored pink. The color coding is used throughout the manuscript. Bound SAM donor substrate is shown in stick format and colored yellow. (**B**) Sequence alignment of human METTL3 and METTL14. The alignment was generated using T-Coffee (*Di Tommaso et al., 2011*) and displayed using Espript (*Gouet et al., 1999*). Invariant residues are highlighted in red; conserved residues are highlighted in yellow. Amino acid residues involved in donor or acceptor substrate binding are marked with colored triangles. Secondary structure elements in METTL3 are depicted above the aligned amino acid sequences. (**C**) Enlarged view of the dimerization interface (orientation as in the left panel of **A**). (**D**) N-terminal helical motif of METTL14 shields the hydrophobic part of the METTL3-METTL14 interface from the solvent exposure by capping it with Ile134[Mettl14]. (**E**) Detailed views of intersubunit interactions mediated by loops Arg468-Lys480[METTL3] and Lys266-Arg283[METTL14] containing hydrophobic residues Trp475-Leu476[METTL3] and Val280-Phe281[METTL14], respectively.

The following figure supplements are available for figure 2:

**Figure supplement 1.** Structural comparisons of the methyltransferase domains of METTL3 and METTL14.

*Figure 2 continued on next page*

eLIFE Research article

*Figure 2 continued*

**Figure supplement 2.** Comparisons of METTL3-METTL14 complex structures.

Lys266-Arg283[METTL14]), that reach over and insert an aromatic residue (Trp475[METTL3] and Phe281-[METTL14], respectively) into a hydrophobic pocket in the other subunit (*Figure 2E*).

The structure of the human METTL3-METTL14 complex has been recently reported by two other studies (*Wang et al., 2016a*, *2016b*). Both studies utilized a slightly different METTL3-METTL14 construct, yielding a different crystal form of the complex. Overall, all structures are in good agreement with one another, the major difference being the absence of an ordered polypeptide chain linking the N-terminal helical motif of METTL14 and the methyltransferase domain in the structure reported in this study (*Figure 2—figure supplement 2*). Additionally, the structure reported in this study lacks a C-terminal helix in METTL14 due to a different choice of METTL14 domain boundaries and instead contains the Cys338[METTL14]–Cys388[METTL14] disulfide bridge. Further structural differences between the different crystal forms can be attributed to the intrinsic flexibility of loop regions and differences in crystal packing (*Figure 2—figure supplement 2*).

Individually, the MTase domains of METTL3 and METTL14 are highly similar to other Rossman fold-containing methyltransferases. The closest structural homologs include bacterial adenine-specific DNA methyltransferases MboIIa (*Osipiuk et al., 2003*) (rmsd of 2.8 Å over 167 Cα atoms for METTL3, 2.7 Å over 165 Cα atoms for METTL14) and RsrI (*Scavetta et al., 2000*) (rmsd of 2.8 Å over 172 Cα atoms for METTL3, 2.8 Å over 171 Cα atoms for METTL14). However, the dimerization mode observed in the METTL3-METTL14 complex is distinct from these other multisubunit RNA methyltransferase enzymes such as Bud23-Trm12 (*Létoquart et al., 2014*) or Trm9-Trm112 (*Létoquart et al., 2015*) and closely resembles the homodimerization mode of prokaryotic N⁶A DNA methyltransferases such as EcoP15I (*Figure 3*) (*Gupta et al., 2015*).

## Donor and acceptor substrate binding in the METTL3 active site

The donor substrate binding site in METTL3 is located at the carboxy-termini of strands β1, β7 and β8, where the SAM donor is bound by multiple hydrogen bonding interactions (*Figure 4A*). The SAM binding site is enclosed from opposite sides by partially disordered loop containing Asp395-

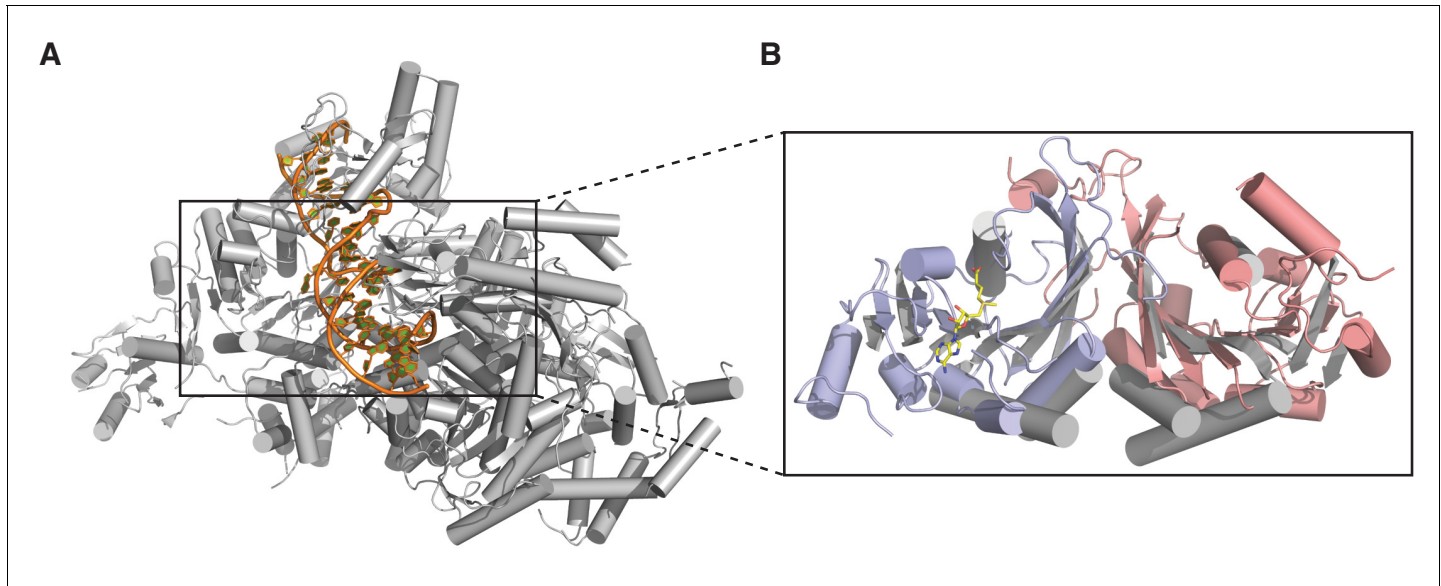

**Figure 3.** Dimerization of METTL3-METTL14 is similar to homodimeric N⁶-adenosine DNA methyltransferases. (**A**) Structure of dimeric N⁶-adenosine DNA methyltransferase EcoP15I bound to a DNA substrate (*Gupta et al., 2015*). (**B**) Inset showing detail of the dimer interface of EcoP15I (colored gray) superimposed with the METTL3-METTL14 (METTL3 colored in blue, METTL14 in pink).

Thr408[METTL3] (termed active site loop 1, ASL1) and a fully ordered loop comprising residues Arg508-Lys513[METTL3] (hereafter termed active site loop 2, ASL2). A conserved DPPW motif (Asp395-Pro398[METTL3]) in ASL1, characteristic of $N^6$-adenosine methyltransferases (*Iyer et al., 2016*), is likely involved in coordinating the adenine group of the acceptor substrate. The methyl group of SAM is positioned directly above the DPPW motif, while the adenine moiety is inserted in a hydrophobic pocket lined with Ile378[METTL3], Pro397[METTL3], Leu409[METTL3] and Phe534[METTL3], and is coordinated by hydrogen binding interactions to Asp377[METTL3] and the main-chain amide nitrogen of Ile378-[METTL3]. The ribose 2' and 3' hydroxyl groups are hydrogen bonded to Asn549[METTL3] and Gln550[METTL3], respectively. The amino group of the methionine moiety is hydrogen bonded to Arg536, while the carboxyl group is surrounded by the side chains of His512[METTL3], Lys513[METTL3], Arg536[METTL3], and His538[METTL3]. Comparisons with the structure of the SAH product-bound complex reveal that the both SAM and SAH molecules adopt very similar conformations except for a slight rotation of the ribose moiety in SAH (*Figure 4B*). Additional ordering of ASL1 is observed in the SAH-bound structure, whereby Tyr406[METTL3] caps the SAH product and makes a hydrogen bonding interaction with Ser511[METTL3] in ASL2. Taken together, the structures of the SAM- and SAH-bound METTL3-14 complexes thus reveal the METTL3 active site in the pre- and post-catalytic states, respectively.

Superposition of METTL3 onto the structure of other methyltransferase-substrate complexes such as EcoP15I suggests a mechanism for acceptor substrate recognition, in which the adenine moiety of the acceptor substrate is inserted between ASL1 and ASL2 (*Figure 4C*). The $N^6$ amino group is located such that it is hydrogen-bonded to the side chain amide group of Asp395[METTL3], the hydroxyl group of Tyr406[METTL3] and the main chain carbonyl of Pro396[METTL3] within the conserved DPPW motif. These interactions activate the amino group for nucleophilic attack on the methyl group in SAM. Additionally, a rearrangement of the ASL1, such that Trp398[METTL3] forms a π-π stacking interaction with the acceptor adenine, would be expected to occur upon acceptor substrate binding.

## METTL14 contains a degenerate active site and lacks catalytic activity

In comparison with METTL3, the MTase domain of METTL14 adopts a very similar fold, superimposing with an rmsd of 1.65 Å over 199 aligned residues (*Figure 2B*, *Figure 2—figure supplement 1*). However, neither SAM nor SAH were observed to be bound to METTL14 in the METTL3-14 complex structures. Modeling a molecule of SAM in the METTL14 active site reveals that the binding would be precluded by steric clashes between the adenine moiety and the side chains of Trp211[METTL14] and Pro362[METTL14] in the absence of compensatory structural changes in METTL14. Moreover, METTL14 lacks residues that could form hydrogen-bonding contacts to the ribose hydroxyls in SAM, as residues Asn549[METTL3] and Gln550[METTL3] correspond to Pro362[METTL14] and Thr363[METTL14], respectively. Additionally, the putative acceptor binding site in METTL14 is occluded by the side chains of Tyr198[METTL14], Tyr199[METTL14] and Ile324[METTL14] projecting from loops corresponding to ASL1 and ASL2 in METTL3 (Glu192-Cys210[METTL14] and Pro319-Lys326[METTL14], respectively) (*Figure 5A*). The equivalent of the METTL3 DPPW motif, Glu192-Leu1195[METTL14] (EPPL), is thus not accessible and moreover lacks an aromatic moiety required for the stacking interaction with the acceptor adenine, (*Figure 5A*). Collectively, these observations suggest that METTL14 contains a degenerate active site that is unable to accommodate donor and acceptor substrates, implying that METTL14 is catalytically inactive. To test this hypothesis, we targeted putative active site residues implicated in donor substrate binding and catalysis in METTL3 and METTL14 by site-directed mutagenesis, and assayed the methyltransferase activities of resulting complexes using an RNA oligonucleotide containing the consensus $m^6A$ methylation sequence GGACU. Substitutions of Asp395[METTL3] or Asn549/Gln550[METTL3] to alanine reduced the methyltransferase activity of the complex to background levels (*Figure 5B*). In contrast, alanine substitutions of Glu192[METTL14] or Pro362/Thr363[METTL14] targeting the putative active site in METTL14 had little effect on the MTase activity of the METTL3-14 complex (*Figure 5B*). These results indicate that the catalytic activity of the complex is restricted to the METTL3 active site and suggest that METTL14 likely plays a non-catalytic role within the $m^6A$ writer complex.

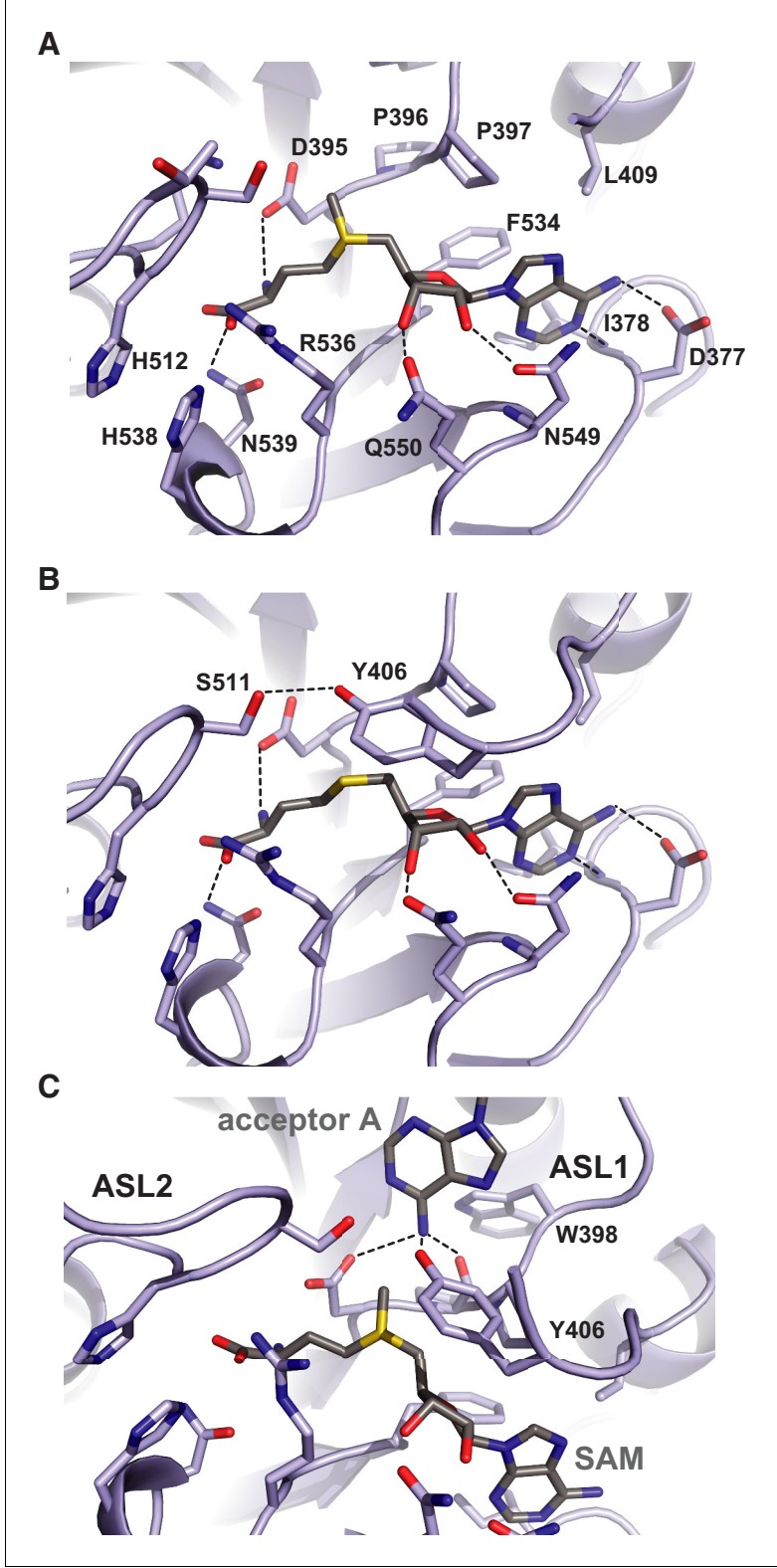

**Figure 4.** Methyl group donor substrate and product binding in the METTL3 active site. The active site of METTL3 with (**A**) the donor substrate SAM and (**B**) product SAH bound. The ligands are shown in stick format and colored yellow. Black dashed lines indicate hydrogen bonding interactions. (**C**) Model of acceptor adenine binding in the METTL3 active site. Donor (SAM) and acceptor (adenine) substrates are shown in stick format and colored gray.

*Figure 4 continued on next page*

*Figure 4 continued*

The model was generated by superimposing METTL3 with the structure of N⁶-adenosine DNA methyltransferases EcoP15I (*Gupta et al., 2015*).

## Substrate RNA binding by the METTL3-METTL14 dimer

Superimposing the structure of the METTL3-14 complex with structures of EcoP15I and TaqI MTases bound to DNA substrates (*Goedecke et al., 2001*; *Gupta et al., 2015*) suggests two potential binding modes for the RNA acceptor substrate, depending on whether the acceptor adenine base is superimposed onto the active site in METTL3 or the corresponding site in METTL14 (*Figure 6A,B*). With the acceptor adenine bound in the active site of METTL3, the ribose-phosphate backbone of the RNA substrate is placed over a positively charged groove located at the interface of the METTL3 and METTL14 subunits (*Figure 6C,D*). This groove is highly evolutionarily conserved and comprises residues Arg465[METTL3], Arg468[METTL3], His474[METTL3], His478[METTL3] and Arg245[METTL14], Arg249-[METTL14], Arg254[METTL14] and Arg255[METTL14]. The last four residues form a positively charged pocket occupied by a co-crystallized acetate ion, possibly mimicking RNA phosphate group binding (*Figure 6E*). These observations suggest that this surface mediates substrate RNA binding and are in good agreement with the presence of bound donor substrate and product in the METTL3 active site and lack of catalytic activity in active site mutants of METTL3. In contrast, the corresponding surface adjacent to the putative active site in METTL14 on the opposite face of the METTL3-14 dimer lacks evolutionary conservation and is mostly negatively charged (*Figure 6D*). This argues against substrate RNA binding by this surface and further underscores the notion that METTL14 is catalytically inactive. To test the hypothesis that the positively charged groove adjacent to the METTL3 active site at the METT3-METTL14 dimer interface is involved in substrate RNA binding, we substituted Arg254[METTL14] with alanine and assayed the resulting complex for MTase activity. The activity of the R254A[METTL14] mutant complex was substantially lower than in the wild type (*Figure 6F*). When Arg255[METTL14] was additionally mutated to alanine, the catalytic activity of resulting R254A/R255A[METTL14] double mutant complex was reduced to background levels, further suggesting that these residues are involved in substrate RNA binding (*Figure 6G*). We additionally tested the involvement of the amino acid residues in the putative acceptor binding pockets in METTL3 and

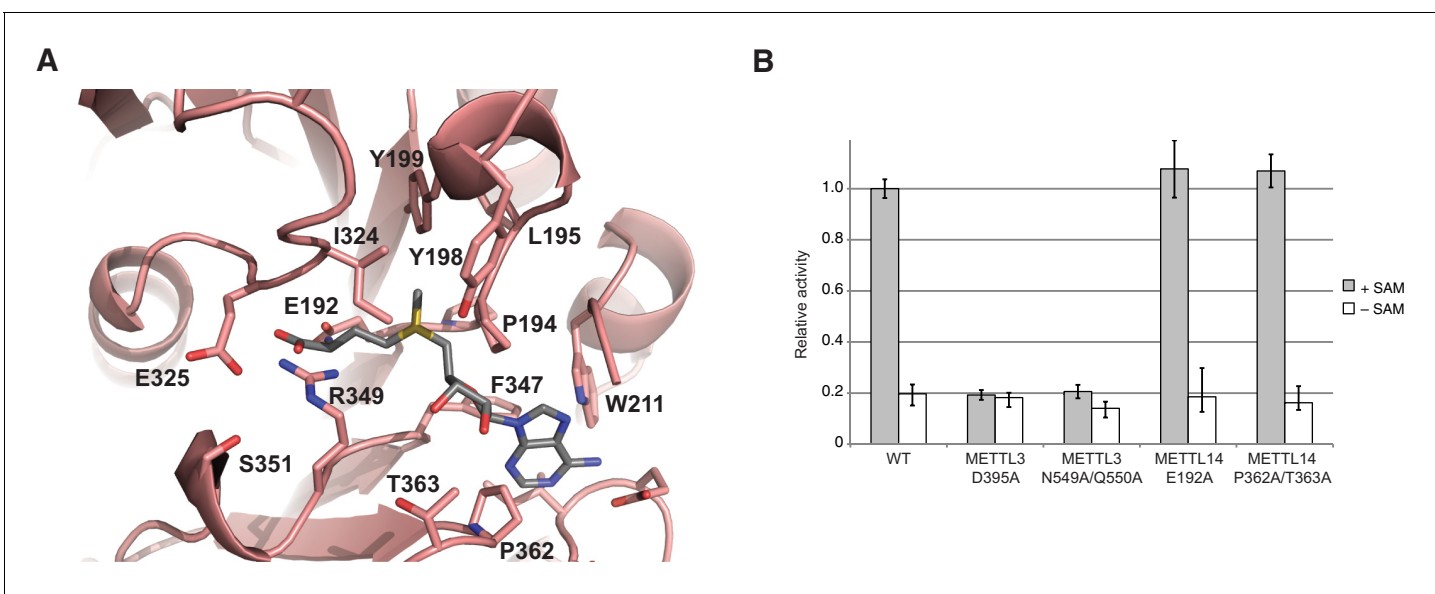

**Figure 5.** METTL14 contains a degenerate active site and is catalytically inactive. (**A**) Zoom-in view of the putative methyltransferase active site in METTL14. The binding of the SAM donor cofactor (shown in grey) was modeled by superimposing the structure of SAM-bound METTL3. (**B**) Methyltransferase activities of the wild-type (WT) METTL3-METTL14 complex and active site alanine mutants.

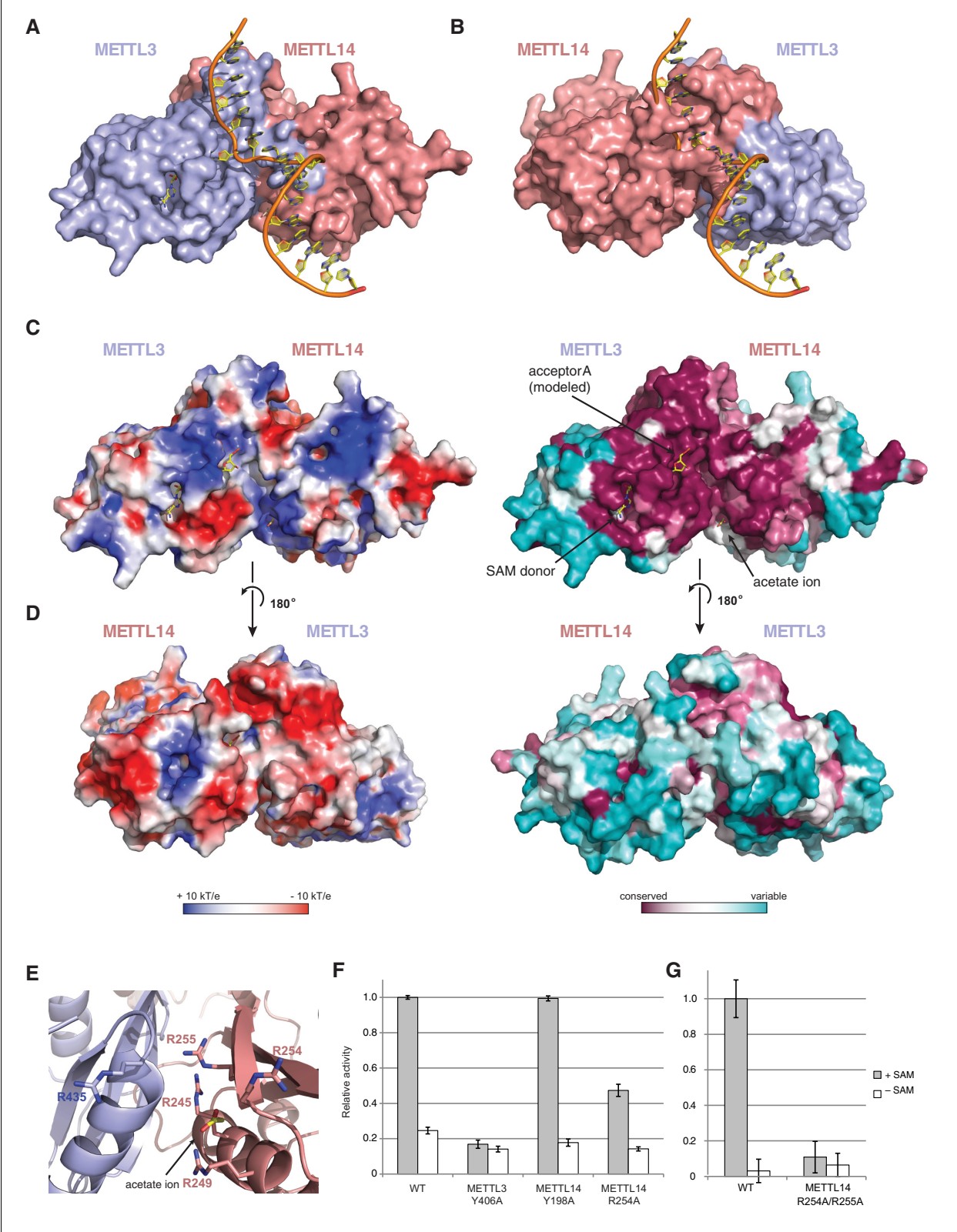

**Figure 6.** The METTL3-METTL14 dimer interface binds substrate RNA. (**A**) Model of acceptor substrate binding in the active site of METTL3 obtained by superimposing METTL3 in the METTL3-METTL14 complex structure with the active methyltransferase subunit in the structure of the Mod2Res1 holocomplex of the N⁶-adenosine DNA methyltransferase EcoP15I bound to substrate dsDNA (PDB entry 4ZCF). For clarity, only the acceptor adenosine-containing strand of the substrate DNA is shown. METTL3-METTL14 is shown in surface representation with subunits colored as in

*Figure 6 continued on next page*

Figure 6 continued

*Figure 2A*. (B) Model of acceptor substrate binding to METTL14 obtained by superimposing METTL14 in the METTL3-METTL14 complex structure with the active methyltransferase subunit in the EcoP15I-dsDNA substrate complex. (C) Surface electrostatic potential (left) and sequence conservation (right) shown for the side of the METTL3-METTL14 heterodimer containing the METTL3 active site. The acetate ion and SAM cofactor co-crystallized with the protein, as well as the acceptor adenosine modeled based on EcoP15I structure are shown in yellow. The surface electrostatic potential was calculated in Pymol (Schroedinger); sequence conservation was mapped using the Consurf server (*Ashkenazy et al., 2010*). (D) Electrostatic surface potential and sequence conservation for the side of the complex corresponding to the degenerate METTL14 catalytic site. (E) Close-up view of the arginine-rich positively charged acetate ion binding site, proposed to interact with the ribose-phosphate backbone of substrate RNA. (F) Methyltransferase activities of WT METTL3-METTL14 complex and the alanine mutants of putative RNA substrate binding residues. The activities were measured as in *Figure 1B*. (G) Methyltransferase activities of WT and R254A/R255A$^{METTL14}$ mutant complex.

METTL14. Alanine substitution of Tyr406$^{METTL3}$ reduced the MTase activity of the complex to background level. In contrast, substitution of Tyr198$^{METTL14}$ had little effect on MTase activity (*Figure 6F*). Collectively, these results suggest that the m$^6$A writer complex contains only one active site – in METTL3 – and that substrate RNA binding is mediated by a conserved electropositive surface at the interface of the METTL3-14 subunits.

## Discussion

Numerous recent studies have established that m$^6$A plays important regulatory roles in eukaryotic RNA metabolism that impact many aspects of cellular physiology. Despite rapid progress in understanding the biological functions of m$^6$A, there have been only limited mechanistic insights into the basic molecular machinery that installs this epitranscriptomic mark. In this study, we have defined the catalytic core of the m$^6$A writer complex and determined the structure of its complexes with methyl group donor substrates and products. Our findings are in good agreement with other recent structural studies of the m$^6$A writer complex (*Wang et al., 2016a*, *2016b*). The molecular architecture of the METTL3-METTL14 dimer is strikingly similar to that of homodimeric bacterial N$^6$-adenosine DNA methyltransferases such as EcoP15I. This hints at an evolutionary scenario in which the m6A writer enzyme emerged by the acquisition of a bacterial homodimeric DNA methyltransferase. Subsequent gene duplication and sequence divergence of the METTL3 and METTL14 subunits may have driven the specificity switch towards RNA.

The presence of an extensive interaction interface between METTL3 and METTL14 in the complex suggests that the two methyltransferase subunits form an obligate, constitutive heterodimer, implying that they are unlikely to be functional in isolation. Of note, we have not been able to express and purify soluble METTL14 in the absence of METTL3, suggesting that METTL3 is required to stabilize METTL14. Our observations are fully consistent with a recent genetic study showing that knockout of either METTL3 or METTL14 in mouse embryonic stem cells resulted in complete loss of m$^6$A modification in mRNA (*Geula et al., 2015*). Although both METTL3 and METTL14 have previously been reported to harbor weak m$^6$A MTase activity (*Liu et al., 2014*), our structural and biochemical data indicate that METTL3 is the sole catalytic subunit within the m$^6$A writer complex. Neither SAM nor SAH was observed to bind to METTL14 in the structures. Moreover, the putative active site in METTL14 is sterically blocked and lacks specific amino acid residues that would facilitate donor and acceptor substrate binding. Consistently, engineered mutations in the active site of METTL3 compromise the MTase activity of the complex, while corresponding mutations in METTL14 have little or no effect. These observations suggest that METTL14 is not catalytically active and likely has a twofold function in maintaining the integrity of the complex and in facilitating RNA substrate binding, explaining the synergistic effect of the METTL3-METTL14 interaction on enzymatic activity of the m$^6$A writer complex.

Based on these results, we propose a model for substrate RNA recognition by the METTL3-METTL14 complex in which binding of the substrate RNA to a conserved, positively charged interface of the METTL3-METTL4 heterodimer positions the acceptor adenine moiety in the catalytic pocket of METTL3 (*Figure 7*). Consistent with the model, mutations of arginine residues at the METTL3-METTL14 interface substantially reduce the MTase activity of the complex. However, given that the truncated complex used for crystallographic analysis lacks catalytic activity in vitro compared

to the complexes containing full-length METTL3, the N-terminal extension of METTL3 is also required for the MTase activity of the core m[6]A complex. It is tempting to speculate that the two N-terminal CCCH zinc finger motifs of METTL3 are essential for catalytic activity, perhaps by contributing to specific recognition of the consensus methylation sequence. Interestingly, despite the spatial proximity of the N-terminal extension of METTL14 to the active site of METTL3 (*Figure 2A*), additional sequences in the N-terminal low complexity region of METTL14 are dispensable for catalytic activity.

Our results provide a revised model of the molecular mechanism of the METTL3-METTL14 complex that extends beyond the recent structural findings of *Wang et al. (2016b)*, showing that the crystallized construct is not representative of the catalytically active complex and implicating the N-terminal region of METTL3 in substrate RNA recognition. Another structural analysis of the METTL3-METTL14 complex, published while this work was under revision (*Wang et al., 2016a*), has corroborated our model and validated the requirement for the METTL3 zinc finger motifs for methyltransferase activity. Nevertheless, the molecular mechanism of substrate RNA recognition still awaits further structural and biochemical characterization. Moreover, the core m[6]A writer complex associates in vivo with additional components such as WTAP, whose functions are not fully understood. Defining the molecular architecture of the m[6]A writer holocomplex will therefore be a major goal of future structural studies.

## Materials and methods

### Protein expression and purification

For biochemical assays, full-length human METTL3-METTL14 complex was prepared by coexpression using the Sf9 insect cell/baculovirus expression system. A two-cassette baculovirus shuttle vector was constructed from plasmid 5B (developed by Scott Gradia, UC Berkeley MacroLab, Addgene #30122) using ligation-independent cloning (LIC). METTL14 was expressed with an N-terminal hexahistidine tag. METTL3 was fused to an N-terminal polypeptide sequence comprising a StrepII epitope tag (WSHPQFEK) fused to green fluorescent protein (GFP). This METTL14 fusion construct was generated by first inserting the METTL14 DNA sequence into the 438-RGFP vector (developed by Scott Gradia, UC Berkeley MacroLab, Addgene #55221) and then subcloning the fusion construct into the 5B baculovirus shuttle vector. Recombinant baculoviruses were generated using the Bac-to-

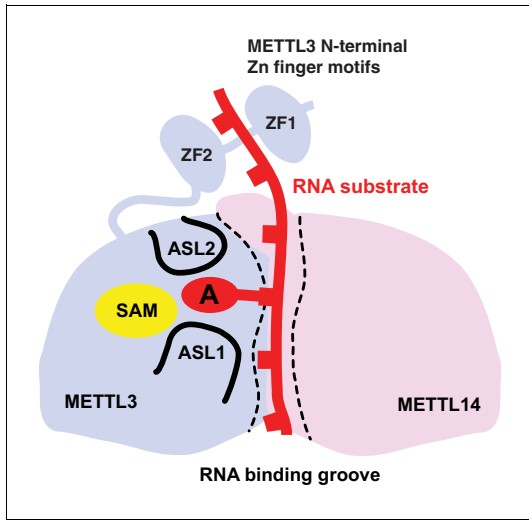

**Figure 7.** Proposed model for RNA methylation by the METTL3-METTL14 complex. Binding of the substrate RNA to a conserved positively charged surface on the asymmetric interface of the METTL3-METTL4 heterodimer positions the acceptor adenine moiety in the catalytic pocket of METTL3 located defined by ASL1 and ASL2 loops. METTL3 catalyzes methyl group transfer from SAM to the acceptor adenine moiety. The N-terminal region in of METTL3, containing two putative CCCH zinc finger motifs, contributes to substrate RNA recognition.

Bac kit (Invitrogen) according to manufacturer's protocol. For protein expression, suspension cultures of Sf9 cells in SF-4 Baculo Express medium (Amimed) were infected at a density of $2.0 \times 10^6$ ml$^{-1}$. Cells were harvested 72 hr post infection, resuspended in 20 mM Tris-Cl pH 8.0, 200 mM KCl, 5 mM imidazole, 10% glycerol, 0.4% Triton-X100, supplemented with Protease Inhibitor Cocktail (Roche Diagnostics GmbH, Germany), and lysed by sonication. The protein complex was purified by Ni-affinity chromatography on Ni-NTA matrix (Qiagen). The affinity tags were removed by digestion with TEV protease, followed by further purification by size exclusion chromatography using the Superdex-200 column (GE Healthcare) in 20 mM Tris-Cl pH 8.0 and 200 mM KCl. Protein samples were concentrated to 2–14 mg ml$^{-1}$ and flash-frozen in liquid nitrogen. Point mutants were generated in the baculovirus shuttle vectors using the QuikChange site-directed mutagesis protocol. Mutant proteins were expressed and as for the WT construct.

For crystallization, the truncated complex containing METTL3 (residues 354–580) and METTL14 (residues 107–395) was prepared by coexpression in Sf9 cells using the 5B baculovirus shuttle vector. In this case, METTL14 carried an N-terminal hexahistidine tag while METTL3 was expressed without an affinity tag. The truncated complex was expressed and purified as for full-length METTL3-METTL14. The sample was concentrated to 11 mg ml$^{-1}$ and flash-frozen in liquid nitrogen. The partially truncated complexes comprising full-length METTL3 and truncated METTL14 (residues 107–395), or truncated METTL3 (residues 354–580) and full-length METTL14, were prepared according to the same procedure.

## Limited proteolysis

The full length METTL3-METTL14 complex was diluted to 1.0 mg ml$^{-1}$ and incubated overnight at 4°C with chymotrypsin at a ratio of 100:1 (w/w). The resulting protein sample was analyzed by SDS-PAGE and subjected to size exclusion chromatography using the Superdex-200 column and 20 mM TRIS pH 8.0, 200 mM KCl buffer. The fractions containing the truncated writer complex were pooled, concentrated and subjected to mass spectrometry and N-terminal Edman sequencing analysis at Functional Genomics Center Zurich.

## Crystallization and structure determination

Purified METTL3$^{354–580}$-METTL14$^{106–396}$ complex was diluted to 5 mg ml$^{-1}$ in 10 mM Tris-Cl pH 8.0, 200 mM KCl. Crystals were obtained using the hanging drop vapor diffusion method by mixing 1 µl complex solution with 1 µl reservoir solution containing 100 mM sodium cacodylate pH 6.5, 100 mM Mg-acetate, 19% PEG 8000 (w/v). For data collection, crystals were cryoprotected by transfer to a solution containing 100 mM sodium cacodylate pH 6.5, 100 mM Mg acetate, 20% (w/v) PEG 8000, and 15% (v/v) glycerol, and flash cooled in liquid nitrogen. X-ray diffraction data were collected at beam line X06DA (PXIII) of the Swiss Light Source (Paul Scherrer Institute, Villigen, Switzerland) and processed using XDS (*Kabsch, 2010*). Data collection statistics are shown in *Table 1*. Native sulfur-SAD data comprised 11 data sets collected by exposing different parts of the same crystal, rotating the crystal through 360° in each data set and changing the kappa angle between datasets in 5° increments to ensure data completeness and redundancy. Sulfur sites were located using SHELXD (*Schneider and Sheldrick, 2002*), and phases were calculated using SHELXE (*Sheldrick, 2008*). The model was subsequently improved using the phenix.autobuild routine (*Zwart et al., 2008*), followed by several rounds of manual building in Coot (*Emsley and Cowtan, 2004*) and refinement using phenix.refine (*Afonine et al., 2012*). Crystals used for the high-resolution native dataset collection were grown in the presence 0.5 mM SAH and 1 mM RNA oligonucleotide (CUGG-m$^6$A-CUAA) or with 5 mM SAM, respectively, from 18–22% (w/v) PEG 3350 and 300 mM Mg-acetate. Substrate-bound structures were determined by molecular replacement using the Phaser module of the Phenix package. Model building and refinement was carried out using COOT and phenix.refine.

## Methyltransferase activity assays

Methyltransferase activities of wild-type and mutant METTL3-METTL14 complexes were measured using an antibody-based assay. Assay conditions were derived from a previously reported radioactivity-based assay (*Li et al., 2016*) and modified to ensure optimal turnover within the timescale of the assay. Reactions mixtures contained 200 nM 3'-biotinylated RNA substrate (5'-UACACUCGAUC UGGACUAAGCUGCUC-3') and 1 µM S-adenosyl methionine in 20 mM Tris buffer at pH 8.0. The

reactions were initiated by the addition of 100 nM protein and incubated at 37°C for 180 min. To detect the production of methylated RNA, reaction mixtures were transferred to the 96-well neutra-vidin coated plates (Pierce) and incubated for 60 min at 4°C. Following extensive washing and block-ing, the plate was incubated first with a $m^6A$-specific primary antibody (Abcam ab151230, used at 1:400 dilution), and subsequently with horseradish peroxidase-conjugated secondary antibody (Sigma-Aldrich, A6154, used at 1:5000 dilution). $m^6A$ antibody binding was quantified by measuring the conversion of a colorimetric substrate 3,3',5,5'-tetramethylbenzidine (TMB, supplied as BM Blue POD substrate by Roche Diagnostics GmbH, Germany) at a wavelength of 390 nm. Reactions lacking the SAM cofactor were performed to measure the background signal for each enzyme construct due to non-specific antibody binding.

## Acknowledgements

We are grateful to Meitian Wang, Vincent Olieric and Takashi Tomizaki at the Swiss Light Source (Paul Scherrer Institute, Villigen, Switzerland) for assistance with X-ray diffraction measurements, and to Serge Chesnov and René Brunisholz (Functional Genomics Center Zurich) for assistance with mass spectrometry and Edman sequencing. We thank members of the Jinek group and Ramesh Pillai for discussions and critical reading of the manuscript. This work was supported by start-up funds from the University of Zurich and the European Research Council (ERC) Starting Grant ANTIVIRNA (Grant no. ERC-StG-337284).

## Additional information

### Funding

| Funder | Grant reference number | Author |
|---|---|---|
| European Research Council | ERC-StG-337284 | Paweł Śledź<br>Martin Jinek |
| Universität Zürich | | Paweł Śledź<br>Martin Jinek |

The funders had no role in study design, data collection and interpretation, or the decision to submit the work for publication.

### Author contributions

PŚ, Wrote the manuscript, Prepared and crystallized the METTL3-14 complex, Collected X-ray data, Determined crystal structures and carried out enzymatic activity assays, Designed experiments, Acquisition of data, Analysis and interpretation of data; MJ, Wrote the manuscript, Assisted with X-ray structure determination and supervised the project, Designed experiments, Acquisition of data, Analysis and interpretation of data

### Author ORCIDs

Paweł Śledź, http://orcid.org/0000-0002-4440-3253
Martin Jinek, http://orcid.org/0000-0002-7601-210X

## Additional files

### Major datasets

The following datasets were generated:

| Author(s) | Year | Dataset title | Dataset URL | Database, license, and accessibility information |
|---|---|---|---|---|
| Sledz P, Jinek M | 2016 | Crystal structure of the human METTL3-METTL14 complex bound to SAH | http://www.rcsb.org/pdb/search/structid-Search.do?structureId=5L6D | Publicly available at the RCSB Protein Data Bank (accession no: 5L6D) |

| Sledz P, Jinek M | 2016 | Crystal structure of the human METTL3-METTL14 complex bound to SAM | http://www.rcsb.org/pdb/search/structid-Search.do?structureId=5L6E | Publicly available at the RCSB Protein Data Bank (accession no: 5L6E) |
|---|---|---|---|---|

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
