## [Decision Letter]

Thank you for submitting your article "Structural insights into the molecular mechanism of the m^6^A writer complex" for consideration by *eLife*. Your article has been reviewed by three peer reviewers, including Jacob Hanna (Reviewer #1), and the evaluation has been overseen by Timothy Nilsen as the Reviewing Editor and James Manley as the Senior Editor.

The reviewers have discussed the reviews with one another and the Reviewing Editor has drafted this decision to help you prepare a revised submission.

As you will see all thought that the work was very high quality and was in principle suitable for *eLife*. While the publication of a similar structure in Nature does not preclude publication of your work in *eLife*, we would like you to address two issues via revision. First, please do a structural superposition and compare conformation of important loop regions. Second make explicit in the text your evidence for "a revised mechanism for substrate RNA recognition by the m^6^A write complex" as stated in your cover letter.

*Reviewer #1:*

This is a fantastic and important study providing 1.9Å crystal structure of METTL3-METTL14 heterodimer complex that is responsible for depositing m^6^A modification on mRNA. The authors establish that Mettl3-Mettl14 hetero-complex forms an asymmetric dimer, with a central RNA binding fold. By exacting mutagenesis of Mett3 and Mettl14 they establish that only METTL3 has the catalytic activity, while METTL14 is still essential for m^6^A complex enzymatic activity through stabilizing the RNA binding and recognition group by the assembled complex.

The manuscript is well written. The Introduction is accurate and informative, references are balanced throughout the manuscript. The figures are elegant and clear. All conclusions made are fully supported by the experimental data.

I only have minor non-experimental revisions suggestions for the authors:

1) Only less than two weeks ago, Wang et al. Nature 2016 published a similar study and reached identical conclusions. Considering the very recent publication, I do not find that the novelty of the current study is compromised. Findings between both studies are totally consistent which is great. I only think the authors should reference Wang et al. Nature 2016 paper.

2) Geula et al. Science 2015 (particularly in the supplementary material section), showed that knockout of Mettl3 or Mettl14 in mouse ESC both resulted in complete loss of m^6^A on mRNA as determined by a variety of methods including MS. The latter occurs despite the presence of Mettl14 upon ablation of Mettl3 protein and vice versa. This genetic result is consistent with conclusions made based on the structural and biochemical in vitro analysis conducted herein, and that both proteins are required for maintaining basal m^6^A deposition by this complex. This should be clearly indicated and discussed in detail.

3) At the end of the Discussion, the authors should indicate that also conducting structural analysis for the larger complex containing either WDDR5, and/or KIAA1429 might be of major importance.

*Reviewer #2:*

This paper by Sledz and Jinek describes the crystal structure of a truncated and inactive form of human METTL3-METTL14 methyltransferase complex, which is responsible in the methylation of adenosine at N6 position (m^6^A) on mRNAs. This is a very interesting field and many high-impact studies have focused on this m^6^A post-transcriptional modification, showing that this dynamic modification affects mRNA fates such as splicing, translation and decay. This modification also influences stem cell differentiation, circadian clock and embryonic development.

Beyond describing the crystal structure of this heterodimeric methyltransferase, the authors show that although both proteins are structurally related to SAM-dependent methyltransferases, only METTL3 binds SAM and is then the catalytic subunit of this holoenzyme, contrary to previously published data. They also identify regions from METTL14, which are essential for enzymatic activity. By comparing their structure to those of DNA m^6^A methyltransferase bound to DNA, they propose that METTL14 contributes to mRNA recognition. This article describes a very nice piece of data on a highly interesting complex. This work is technically of great quality and the results are of interests.

The main concern about this work is that as mentioned in the manuscript, the crystal structure of the same human METTL3-METTL14 complex has just been published in Nature (Wang et al; Nature; 2016). As the boundaries of the crystallized complex are quite similar to those used in the Nature's study, the current structure does not bring much additional information. One aspect, which could be interesting, would be to compare these structures as these complexes did not crystallized in the same space group. Hence, some interesting differences might exist. Similarly, the biochemical data presented in this manuscript do not bring much novelty compared to those presented in the Nature's paper.

In conclusion, this work would have been very interesting but as another group recently published similar structure, there is no more novelty.

Reviewer #3:

m^6^A methylation of adenosine is a prevalent and reversible modification found in eukaryotic messenger RNA and long non-coding RNAs. This modification is not only important for its role in regulation of stability, translation and processing of cellular transcripts but is also vital for stem cell differentiation and embryonic development. Its regulatory role is also signified by the discovery of demethylases that reverses this reaction. The forward reaction for m^6^A methylation is catalyzed by a complex composed of two proteins METTL3 and METTL14, which are actively under investigation in recent years. In this manuscript, the authors obtained high resolution crystal structures of a minimal domain constructs of METTL3-METTL14 complex in the presence of the methyl donor substrate (SAM) and the product (SAH). In essence, the structure shows that METTL3 -METTL14 forms a stable complex with extensive hydrophobic and hydrogen bonding interactions. The authors also point out that, although the catalytic site resides in the METTL3, the full length METLL14 is also an essential component for the catalytic activity. They also verified this observation by showing a structure superposition of METTL3-METTL14 complex with the structure of DNA methyltransferase EcoP15I bound to substrate DsDNA. The model suggests that the substrate RNA binds to a conserved positively charged interface of METTL3-METTL14 complex while positioning the acceptor adenine moiety in the catalytic pocket of METTL3.This observation was further investigated by point mutation at the proposed binding surface responsible for the RNA binding.

Technical aspects:

The paper is clearly written and appropriately referenced. The data statistics for the crystal structures are appropriate and refined well. However, the quality of figures needs to be improved before the final publication. For example,

1) The secondary structure motifs in Figure 4 should be labelled. The color code for SAM and SAH in Figure 6 should be changed. This reviewer could hardly see any difference between the color of Carbon and Sulphur (6A, 6B). I propose to apply the same color coding for SAH and SAM in all three figures.

2) In the sequence alignment, please use the real residue numbers rather than the scaling numbers. Also, mark and write the details of the active site residues of METTL3 and METTL4 and also give a description in the footnote. No need to show a consensus sequence of alignment.

Originality and significance:

The crystal structure of METTL3-METTL14 is undoubtedly of broad interest considering the central role of the METTL3-METTL14 complex in mRNA methylation that leads to cellular regulation and human disease and deserves publication in *eLife*. Moreover, the authors contradict the previous reports of presence of activity in the isolated METTL3 and METTL14 subunits. This lack of activity is also explained convincingly by the crystal structure of the complex. METTL3 has been shown to be the catalytic subunit while METTL14 has been shown to be mainly important for RNA recognition and binding. However, it is worth noting that the current work is very similar to the work recently been published by Wang et al. (Nature 2016), which dilutes the uniqueness of the current work. Therefore, I would leave it to the editor's discretion whether the work is deemed novel enough for publication in *eLife*.

The content of the current paper in its present form could be improved to answer a few outstanding questions previously not touched by Wang et al. (see below).

A) One of the conclusions of the manuscript is that the MTase activity of the complex requires the full length METTL14 and at least the catalytic domain of METTL3. This is because the construct METTL14 107-395 does not show any activity when complexed with either the full length or catalytic domain of METTL3 (Figure 1). However, to pin point the important region of METTL14 for the MTase activity, a few more plausible constructs should be designed. I propose to perform activity measurements of METTL3 (356-580) complexed with two more METTL14 constructs lacking either the C-terminal (res 1-395) or the N-terminal region (res 107-456). This could further reveal the substrate recognition and binding regions of the complex. Furthermore, a possible substrate recognition site (that detect the consensus sequence in RNA) should be referred to in Figure 10.

B) The author showed that point mutation of R254A (in METTL14) decreases the MTase activity by around 50 percent (Figure 9F). In my opinion, this is not enough to confirm the RNA binding surface. I would propose to map the binding surface by performing more mutations at the single and at the combination of various interface residues that would probably show background level activity. Other option is to perform EMSA experiment.

---

## [Author Response]

*As you will see all thought that the work was very high quality and was in principle suitable for eLife. While the publication of a similar structure in Nature does not preclude publication of your work in eLife, we would like you to address two issues via revision. First, please do a structural superposition and compare conformation of important loop regions. Second make explicit in the text your evidence for "a revised mechanism for substrate RNA recognition by the m^6^A write complex" as stated in your cover letter.*

As requested by the Reviewing Editor, we have made a structural superposition of our structure and the structure solved by Ping Yin and colleagues (Wang et al., 2016b) and make comparisons of the structures in the Results section (subsection “METTL3 and METTL14 form a pseudosymmetric dimer”, second paragraph and Figure 2—figure supplement 2).

We have also revised the Discussion section in order to make an explicit mention for a revised model for the molecular mechanism of m^6^A modification by the METTL3-METTL14 complex. In our Discussion, we now state:

“Our results provide a revised model of the molecular mechanism of the METTL3-METTL14 complex that extends beyond the recent structural findings of Wang et al. (Wang et al., 2016b), showing that the crystallized construct is not representative of the catalytically active complex and implicating the N-terminal region of METTL3 in substrate RNA recognition. Another structural analysis of the METTL3-METTL14 complex, published while this work was under revision (Wang et al., 2016a), has corroborated our model and validated the requirement for the METTL3 zinc finger motifs for methyltransferase activity.”

*Reviewer #1:*

*[…] I only have minor non-experimental revisions suggestions for the authors:*

*1) Only less than two weeks ago, Wang et al. Nature 2016 published a similar study and reached identical conclusions. Considering the very recent publication, I do not find that the novelty of the current study is compromised. Findings between both studies are totally consistent which is great. I only think the authors should reference Wang et al. Nature 2016 paper.*

We now refer to both the Nature paper of Ping Yin and colleagues (Wang et al., 2016b) as well as the Molecular Cell paper by Yunsum Nam and colleagues (Wang et al., 2016a), which has been published while this manuscript was in revision. Both studies are now mentioned in the Results section (subsection “METTL3 and METTL14 form a pseudosymmetric dimer”, second paragraph) and in the Discussion (first and last paragraphs). As requested by the Reviewing Editor, we have provided a comparison of our structure and that of Yin and colleagues in Figure 2—figure supplement 2.

*2) Geula et al. Science 2015 (particularly in the supplementary material section), showed that knockout of Mettl3 or Mettl14 in mouse ESC both resulted in complete loss of m^6^A on mRNA as determined by a variety of methods including MS. The latter occurs despite the presence of Mettl14 upon ablation of Mettl3 protein and vice versa. This genetic result is consistent with conclusions made based on the structural and biochemical in vitro analysis conducted herein, and that both proteins are required for maintaining basal m^6^A deposition by this complex. This should be clearly indicated and discussed in detail.*

We agree with the reviewer that this experimental observation deserves a greater mention in the Discussion. We have accordingly revised the Discussion section to state the following:

“The presence of an extensive interaction interface between METTL3 and METTL14 in the complex suggests that the two methyltransferase subunits form an obligate, constitutive heterodimer, implying that they are unlikely to be functional in isolation. […] Our observations are fully consistent with a recent genetic study showing that knockout of either METTL3 or METTL14 subunits in mouse embryonic stem cells resulted in complete loss of m^6^A modification in mRNA (Geula et al., 2015).”

*3) At the end of the Discussion, the authors should indicate that also conducting structural analysis for the larger complex containing either WDDR5, and/or KIAA1429 might be of major importance.*

We are in full agreement with the reviewer concerning the importance of further structural studies of the m^6^A writer holocomplex. As suggested by the reviewer, we have revised the Discussion to include the following concluding statements:

“Moreover, the core m^6^A writer complex associates in vivo with additional components including WTAP and KIAA1429, whose functions have remained elusive thus far. Defining the molecular architecture of the m^6^A writer holocomplex will therefore be a major goal of future structural studies.”

*Reviewer #2:*

*[…] The main concern about this work is that as mentioned in the manuscript, the crystal structure of the same human METTL3-METTL14 complex has just been published in Nature (Wang et al; Nature; 2016). As the boundaries of the crystallized complex are quite similar to those used in the Nature's study, the current structure does not bring much additional information. One aspect, which could be interesting, would be to compare these structures as these complexes did not crystallized in the same space group. Hence, some interesting differences might exist. Similarly, the biochemical data presented in this manuscript do not bring much novelty compared to those presented in the Nature's paper.*

*In conclusion, this work would have been very interesting but as another group recently published similar structure, there is no more novelty.*

We thank the reviewer for the appreciation of our work. As suggested by the reviewer and the Reviewing Editor, we have compared our structure with the structure published by Ping Yin and colleagues (Wang et al., 2016b) as well as the Molecular Cell paper by Yunsum Nam and colleagues (Wang et al., 2016a) in the manuscript. We discuss the differences in the Results section (subsection “METTL3 and METTL14 form a pseudosymmetric dimer”, second paragraph) and in Figure 2—figure supplement 2.

*Reviewer #3:*

*Technical aspects:*

*The paper is clearly written and appropriately referenced. The data statistics for the crystal structures are appropriate and refined well. However, the quality of figures needs to be improved before the final publication. For example,*

*1) The secondary structure motifs in Figure 4 should be labelled. The color code for SAM and SAH in Figure 6 should be changed. This reviewer could hardly see any difference between the color of Carbon and Sulphur (6A, 6B). I propose to apply the same color coding for SAH and SAM in all three figures.*

We have labeled the secondary structure motifs in Figure 2 and Figure 2—figure supplement 1.

We have also changed the color code of the SAM/SAH ligands in Figure 4 (formerly Figure 6) to distinguish better the carbon atoms from the sulfur atom in the ligand.

*2) In the sequence alignment, please use the real residue numbers rather than the scaling numbers. Also, mark and write the details of the active site residues of METTL3 and METTL4 and also give a description in the footnote. No need to show a consensus sequence of alignment.*

We have revised the alignment (now in Figure 2) to include the METTL3 and METTL14 sequence numbering. We have also labeled specific amino acid residues involved in donor or acceptor substrate binding and highlight the active site loops. We have included a more detailed description of the alignment in the figure legend.

Originality and significance:

*The crystal structure of METTL3-METTL14 is undoubtedly of broad interest considering the central role of the METTL3-METTL14 complex in mRNA methylation that leads to cellular regulation and human disease and deserves publication in eLife. Moreover, the authors contradict the previous reports of presence of activity in the isolated METTL3 and METTL14 subunits. This lack of activity is also explained convincingly by the crystal structure of the complex. METTL3 has been shown to be the catalytic subunit while METTL14 has been shown to be mainly important for RNA recognition and binding. However, it is worth noting that the current work is very similar to the work recently been published by Wang et al. (Nature 2016), which dilutes the uniqueness of the current work. Therefore, I would leave it to the editor's discretion whether the work is deemed novel enough for publication in eLife.*

*The content of the current paper in its present form could be improved to answer a few outstanding questions previously not touched by Wang et al. (see below).*

We thank the reviewer for the critical assessment of our work.

*A) One of the conclusions of the manuscript is that the MTase activity of the complex requires the full length METTL14 and at least the catalytic domain of METTL3. This is because the construct METTL14 107-395 does not show any activity when complexed with either the full length or catalytic domain of METTL3 (Figure 1). However, to pin point the important region of METTL14 for the MTase activity, a few more plausible constructs should be designed. I propose to perform activity measurements of METTL3 (356-580) complexed with two more METTL14 constructs lacking either the C-terminal (res 1-395) or the N-terminal region (res 107-456). This could further reveal the substrate recognition and binding regions of the complex. Furthermore, a possible substrate recognition site (that detect the consensus sequence in RNA) should be referred to in Figure 10.*

We have revisited the catalytic activities of the truncated constructs of the METTL3-METTL14 complex. We discovered that our samples METTL3(354-580)–METTL14(2-456) and METTL3(2-580)–METTL14(107-395), which had been purified in parallel, had got accidentally swapped post purification.

As a result, the lack of catalytic activity shown in Figure 1 corresponds to the METTL3(354-580)–METTL14(2-456), i.e. the sample in which METTL3 lacks its N-terminal region. This indicates that the N-terminal region containing the two putative zing finger motifs is required for the catalytic activity of the complex, rather than the N- and C-terminal extensions in METTL14, as we previously stated in the original version of the manuscript. We are deeply sorry for this error and apologize for the confusion that we have caused.

Our data now suggests that the N-terminal zinc finger motifs in METTL3 contribute to substrate recognition, possibly by providing sequence specific recognition of the methylation consensus sequence. Based on these results, we have revised our model for the molecular mechanism of the METTL3-METTL14 complex (now shown in Figure 7). Our data is fully consistent with the structural and biochemical study of METTL3-METTL14 by Yunsum Nam and colleagues (Wang et al., 2016a), published in Molecular Cell last week, which arrived at the same conclusion.

*B) The author showed that point mutation of R254A (in METTL14) decreases the MTase activity by around 50 percent (Figure 9F). In my opinion, this is not enough to confirm the RNA binding surface. I would propose to map the binding surface by performing more mutations at the single and at the combination of various interface residues that would probably show background level activity. Other option is to perform EMSA experiment.*

In addition to testing the R254A mutation in METTL14, we have now also tested the R254A/R255A double mutant. The activity of the METTL3-METTL14 complex containing this double mutation was reduced to background level (Figure 6), strongly suggesting that these residues are involved in substrate RNA binding.